# Model-Free Adaptive Control of Second-Order Multi-Agent Systems via Backstepping Under Mixed Attacks

1st Lei Han
*College of Automation*
*Shenyang Aerospace University*
Shenyang, China
h20009262024@163.com

2nd Dong Liu*
*College of Automation*
*Shenyang Aerospace University*
Shenyang, China
sy04848@126.com

*Abstract*—This paper employs model-free adaptive control methods to investigate second-order multi-agent systems under mixed attacks. Based on backstepping techniques, second-order multi-agent systems are equivalently transformed into two interconnected first-order subsystems to reduce structural complexity. Additionally, considering the impacts of multi-source mixed network attacks,compensation mechanism employing predictive distributed output are designed, which mitigate the negative consequences of such attacks. Furthermore, leveraging only input-output data from each subsystem, a distributed model-free adaptive control scheme utilizing backstepping is developed to achieve consensus. Finally, the effectiveness of the proposed method is validated through two simulation examples.

*Index Terms*—mixed attack, data-driven control, second-order multi-agent systems, backstepping method

## I. INTRODUCTION

In recent years, multi-agent systems (MASs) have seen widespread application and emerge as a significant research direction in the field of network intelligence. MASs achieve numerous outstanding research results in various practical fields, such as remote unmanned aerial vehicles (UAVs) [1], multirotor drone systems [2], and multi-robot systems [3]. Compared to traditional control systems, multi-agent systems offer several advantages, including distribution, autonomy, efficiency, and cost-effectiveness. Advantages make multi-agent systems particularly appealing for a wide range of applications, from industrial automation to environmental monitoring, and from defense systems to intelligent transportation networks.

Typically, current research on cooperative control problems can be classified into three primary directions: leaderless cooperative control [4], leader-follower cooperative control [5], and containment control [6]. Leaderless cooperative control focuses on achieving consensus among agents without a designated leader, emphasizing decentralized decision-making and robustness to individual agent failures. Leader-follower cooperative control, involves a hierarchical structure where certain leaders guide the behavior of the followers, facilitating coordinated actions and efficient task completion. Containment control aims at ensuring that certain leaders steer the group of followers to stay within a desired area or follow a specific trajectory, which is crucial in applications such as formation flying and swarm robotics.

It is important to note that cybersecurity plays a crucial role in MASs. In general, network attacks include denial of service (DoS) attacks, false data injection (FDI) attacks, deception attacks (DA), and more. Attacks can have various unexpected impacts on the system, disrupting normal operations and potentially causing significant damage. Consequently, escalating emphasis is placed on researching service denial assaults within academic community. The work [7] introduces a collaborative defense mechanism aimed at countering sporadic incidents of such attacks. In [8], event-triggered state observer is developed to treat attack signals as states. In [9], compensation for attacks is achieved by inversely compensating for deception attacks, mitigating adverse effects on the system.

In [10], predictive control algorithms apply to multi-agent systems with communication constraints. Predictive control algorithms enable MASs to achieve cooperative control even when network attacks are present. By anticipating future states and adjusting control inputs, predictive control enhances resilience to disruptions and uncertainties. When reliable communication cannot be guaranteed, the system must dynamically adapt to the ever-changing environment and potential threats. The integration of predictive control algorithms represents a significant advancement in MASs capability to operate effectively in complex and adversarial environments, ensuring robust performance and maintaining operational goals despite network attacks.

It is worth noting that the majority of current research on MASs under network attacks presupposes a basic premise: the model of the system is known. However, in numerous complex systems, securing an exact model is difficult, which considerably narrows the depth and breadth of cooperative control studies. Yet, the challenge is being addressed with the emergence of data-driven control strategies introduced by Hou et al. [11], which has been further developed in later publications [12]-[13]. Such strategies for control do not rely on the underlying mathematical model of the system; they

function solely with input and output data. This strategy has unveiled opportunities for examining many intricate systems, permitting researchers to delve into cooperative control issues more profoundly. Presently, an extensive collection of studies utilizes the technique to explore cooperative control issues further [14]-[15]. However, when it comes to more complex second-order systems, the technique still grapples with offering a sufficient and holistic solution.

Backstepping is a pivotal control algorithm in the realm of nonlinear control. The specific design process involves breaking down the model of the controlled system into several subsystems to ensure stability. Step-by-step design is then applied to intermediate virtual control variables, followed by monitoring the performance of these virtual variables to achieve desired control objectives. [17] introduces a backstepping strategy for tracking control utilizing state observers to enhance system performance. [18] utilizes the backstepping method with command control to bypass the complexity of intermediate virtual variables, simplifying the control process. Reference [19] proposes a backstepping approach that addresses the computational challenges found in traditional backstepping algorithms. [20] combines backstepping with a model-free approach to solve control problems by dividing second-order systems into two interrelated subsystems. In summary, the combination of backstepping and model-free control effectively tackles tracking control problems under external disturbances, providing robust solutions in complex environments.

Drawing on the findings mentioned earlier, the paper delves into control strategies for second-order nonlinear multi-agent systems, employing a backstepping approach. When compared with existing literature on multi-agent systems, the highlights are:

(1) This paper applies the backstepping method to second-order multi-agent systems, converting the systems into two interconnected serial first-order subsystems. By creating virtual desired velocities, the development of the controller for the second-order system is achieved.

(2) Considering that sensor-controller communication channels suffer from mixed network attacks, we design a compensation mechanism that uses predictive distributed output to counteract the received attacks.

(3) The proposed distributed model-free adaptive control scheme based on backstepping utilizes only real-time measured input-output (I/O) data and information from neighboring nodes, which positions it as the data-driven consensus control method for multi-agent systems.

The subsequent sections will provide an in-depth exposition of the core material of this paper. Section II will delve into the principles of graph theory and the formalization of the issue at hand. Section III presents the controller design for second-order systems considering mixed network attacks and convergence analysis. Section IV provides a numerical case to demonstrate the control effectiveness of the proposed control scheme. Section V concludes the paper.

## II. GRAPH THEORY AND PROBLEM FORMULATION

### A. Graph Theory

The real numbers are represented by the symbol $\mathbb{R}$. For any matrix $A \in \mathbb{R}^{N \times N}$, the norm is indicated as $\|A\|$. The term $\text{Diag}(\cdot)$ is used to denote a matrix with entries only on its main diagonal, while $I$ denotes the standard matrix of the same size with ones on the diagonal and zeros elsewhere. Graph theory serves as a crucial framework in multi-agent systems to model interaction topologies. Consider a weighted directed graph $G = (V, E, A)$, where $V = \{1, 2, \ldots, N\}$ is the set of vertices, $E \subseteq V \times V$ is the set of edges, and $A$ is the matrix that encodes the connections between nodes.. The vertices $V$ also function as indices for the agents.

When agent $j$ receives a message from agent $i$, the edge $(i, j) \in E$ exists. Here, agent $j$ is referred to as the child of agent $i$, and agent $i$ is the parent of agent $j$. The neighborhood of agent $i$ is described as $\mathcal{N}_i = \{j \in V \mid (j, i) \in E\}$. The weighted adjacency matrix $A = (a_{i,j}) \in \mathbb{R}^{N \times N}$ is defined such that $a_{i,i} = 0$ and $a_{i,j} = 1$ if $(j, i) \in E$; otherwise, $a_{i,j} = 0$.

The Laplacian matrix $L$ of graph $G$ is defined as $L = D - A$, where $D = \text{diag}(d_{in,1}, d_{in,2}, \ldots, d_{in,N})$. The in-degree $d_{in,i}$ of vertex $i$ is given by $\sum_{j=1}^{N} a_{i,j}$. A graph is considered connected when it is possible to move from any node to another through a viable route.

### B. Problem Formulation

Let us examine the subsequent second-order nonlinear system in discrete time:

$$
\begin{cases}
y_p(k+1) = h_p(y_p(k), x_p(k)) \\
x_p(k+1) = f_p(x_p(k), u_p(k))
\end{cases}
\tag{1}
$$

in this case, Additionally, $f_p(\cdot)$ and $h_p(\cdot)$ denotes an unknown nonlinear function.

*For subsystem 1*

*Assumption 1* [21]: The subsystem 1 adheres to the generalized Lipschitz condition, indicating that if the variation in control input $\Delta u_p(k-1) \neq 0$, then $\Delta x_p(k) \leq b_1 \Delta u_p(k)$, where $\Delta x_p(k) = x_p(k) - x_p(k-1)$ and $\Delta u_p(k) = u_p(k) - u_p(k-1)$. The constant $b_1$ is positive.

*Assumption 2* [21]: The derivatives of the function $f_p(\cdot)$ concerning the system's control signal $u_p(k)$ are uninterrupted.

*Assumption 3*: There exists an upper limit to the proportion of the variation in $u_q(k)$ relative to $u_p(k)$ is bounded, i.e., $|\Delta u_q(k)/\Delta u_p(k)| < \varsigma$, with $\varsigma$ being a positive constant.

*Assumption 4* [22]: The attribute of directed graph $\mathcal{N}$ is its possession of a directed root-branch structure that spans all its vertices.

*Remark 1:* Assumptions 1 and 2 are commonly found in the domain of regulatory systems and support the application of dynamic linearization techniques throughout the present work. Assumption 3 is employed in the following proof. Assumption 4 is crucial to achieve consensus in leader-follower coordination.

*Lemma 1* [23]: Assuming that subsystem 1 adheres to Assumptions 1 and 2 are satisfied and $|\Delta u_p(k)| \neq 0$ for all time $k$, then a pseudo-partial derivative (PPD) parameter $\Phi_{p1}(k)$ exists. Consequently, the formulation of subsystem 1 can be restructured to reflect this:

$$\Delta x_p(k+1) = \Phi_{p1}(k)\Delta u_p(k) \qquad (2)$$

where $\Phi_{p1}(k)| \leq \bar{\mathfrak{b}}_1$

*Similarly, for subsystem 2*

*Assumption 5* [21]: The subsystem 2 adheres to the generalized Lipschitz condition, indicating that if the variation in control input $\Delta x_p(k-1) \neq 0$, then $\Delta y_p(k) \leq b_2 \Delta x_p(k)$, where $\Delta y_p(k) = y_p(k) - y_p(k-1)$ and $\Delta x_p(k) = x_p(k) - x_p(k-1)$. The constant $b_2$ is positive.

*Assumption 6* [21]: It is assumed that the derivatives of $f_p(\cdot)$ concerning the control input $u_p(k)$ of the system are unbroken throughout their domain.

*Assumption 7*: It is posited that the proportional change in $x_q(k)$ relative to $x_p(k)$ is confined within a limit, i.e., $|\Delta x_q(k)/\Delta x_p(k)| < \zeta_1$ with $\zeta_1$ being a positive constant.

*Assumption 8* [22]: The directed graph $\mathcal{G}$ is characterized by the presence of a directed spanning tree.

*Lemma 2* [23]: For subsystem 2, if Assumptions 5 and 6 are satisfied and $|\Delta x_p(k)| \neq 0$ for all time $k$, then a pseudo-partial derivative (PPD) parameter $\Phi_{p2}(k)$ exists. Consequently, subsystem 2 can be expressed as:

$$\Delta y_p(k+1) = \Phi_{p2}(k)\Delta x_p(k) \qquad (3)$$

where $\Phi_{p2}(k)| \leq \bar{\mathfrak{b}}_2$

For the overall system

$$\Delta y_p(k+1) = \Phi_p(k)\Delta u_p(k-1) \qquad (4)$$

where $\Phi_p(k) = \Phi_{p1}(k-1)\Phi_{p2}(k)$

The error term associated with the output of the $p$th agent, when distributed, is delineated as follows:

$$\varepsilon_p(k+1) = \sum_{p \in \mathcal{N}_p} a_{pq}(y_p(k+1) - y_q(k+1)) \qquad (5)$$

By substituting equation (1) into equation (5) and introducing a new nonlinear function $\mathcal{F}(\cdot)$, equation (5) becomes:

$$\varepsilon_p(k+1) = \mathcal{F}_p(y_p(k), u_p(k), y_q(k), u_q(k)) \qquad (6)$$

*Theorem 1:* For equation (6), if assumptions 1-3 hold true and $|\Delta u_p(k)| \neq 0$ at all times $k$, it follows that there exists a pseudo-partial derivative (PPD) parameter $\Phi_p(k)$. Consequently, the formulation of equation (6) is rephrased to:

$$\Delta \varepsilon_p(k+1) = \Phi_p(k)\Delta u_p(k) \qquad (7)$$

where $\Delta \varepsilon_p(k+1) = \varepsilon_p(k+1) - \varepsilon_p(k)$ and $|\Phi_p(k)| \leq \mathfrak{b}_p$, with $\mathfrak{b}_p$ being a positive constant.

Proof: Utilizing system (6), $\Delta \varepsilon_p(k+1)$ is calculated as

$$\begin{aligned}
\Delta \varepsilon_p(k+1) = &\mathcal{F}_p[y_p(k), u_p(k), y_q(k), u_q(k)] \\
&- \mathcal{F}_p[y_p(k-1), u_p(k-1), y_q(k-1), u_q(k-1)] \\
&+ \mathcal{F}_p[y_p(k-1), u_p(k-1), y_q(k-1), u_q(k-1)] \\
&- \mathcal{F}_p[y_p(k-1), u_p(k-1), y_q(k), u_q(k)]
\end{aligned}$$

Then, utilizing the mean value theorem for differentiation on the difference $\mathcal{F}_p[y_p(k), u_p(k), y_q(k), u_q(k)] - \mathcal{F}_p[y_p(k-1), u_p(k-1), y_q(k), u_q(k)]$ with respect to $u_p(k)$, it is possible to derive

$$\Delta \varepsilon_p(k+1) = \frac{\partial \mathcal{F}_p^*}{\partial u_p(k)}\Delta u_p(k) + \Psi_p(k)$$

where $\left(\frac{\partial \mathcal{F}_p^*}{\partial u_p(k)}\right)$ denotes the partial derivative value of $\mathcal{F}_p$ with respect to $u_p(k)$ in the interval $[u_p(k-1), u_p(k)]$. The term $\Psi_p(k)$ is defined as $\mathcal{F}_p[y_p(k-1), u_p(k-1), y_q(k), u_q(k)] - \mathcal{F}_p[y_p(k-1), y_q(k), u_q(k), u_p(k-1)]$.

Examine the equation for data discrepancies $\Psi_p(k) = \eta_p(k)\Delta u_p(k) + \eta_q(k)\Delta u_q(k)$ with variable $\eta_p(k)$ and $\eta_q(k)$ for every predetermined moment $k$. According to Assumption 3, there exists a solution $\eta_p^*(k)$ where $\Psi_p(k) = \eta_p^*(k)\Delta u_p(k)$ is valid. Letting $\Phi_p(k) = \left(\frac{\partial \mathcal{F}_p^*}{\partial u_p(k)}\right) + \eta_p^*(k)$, the formula above can be derived. Additionally, in accordance with Assumptions 1 and 4, A number of constants exist with the property that $\bar{c}_p > 0$ and $\bar{c}_q > 0$ such that:

$$\begin{aligned}
|\Delta \varepsilon_p(k+1)| \leq& \bar{c}_p a_{pp}|\Delta u_p(k)| + \sum_{q \in \mathcal{N}_p} \bar{b}_q a_{pq}|\Delta u_q(k)| \\
\leq& \bar{c}_p a_{pp}|\Delta u_p(k)| + \sum_{q \in \mathcal{N}_p} \bar{c}_q a_{pq}|\Delta u_q(k)| \\
\leq& \left(\bar{c}_p a_{pp} + \sum_{q \in \mathcal{N}_p} \bar{b}_q a_{pq}\right)|\Delta u_p(k)| \\
=& c_p|\Delta u_p(k)|
\end{aligned}$$

where $c_p = \bar{c}_p a_{pp} + \sum_{q \in \mathcal{N}_p} \bar{b}_q a_{pq}|e|$. Therefore, one can follow from the formula above that the PPD parameter $\Phi_p(k)$ remains bounded, i.e., $|\Phi_p(k)| \leq c_i$.

## III. MAIN RESULTS

This section elaborates on the main content of this article, including the design of mixed network attacks, compensation mechanisms, controller design, and proof of bounded distributed output errors. The theoretical structure of the article is illustrated in Fig.1.

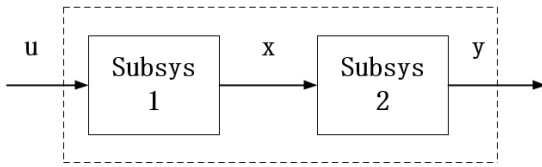

Fig. 1. System model decomposition diagram.

## A. Mixed Cyber-attacks Design

In this section, we will detail the development of a composite cyber-assault strategy. The attack, consisting of DoS, FDI, and DA, constitutes a stochastic attack. $\varepsilon_{pi}(k)$ represents the output signals under different attacks, with $\bar{\varepsilon}_p(k)$ denoting the final output signal subjected to the mixed attack.

In the scenario where $i = 1$, the system endures a specific form of DoS attack, and the corresponding formula, labeled as equation (5), can be articulated as follows:

$$\varepsilon_{p1}(k) = l_1(k)\varepsilon_p(k) \tag{8}$$

where $l_1(k)$ represents the success of the DoS attacks and follows a Bernoulli distribution. If $l_1(k) = 0$, it shows that the assaults achieved their intended outcome, with the likelihood of $\mathbb{P}\{l_1(k) = 0\} = \bar{l}$. Conversely, if $l_1(k) = 1$, the offensives did not succeed, and the likelihood is $\mathbb{P}\{l_1(k) = 1\} = 1 - \bar{l}$.

In the case where $i = 2$, the system experiences FDI attacks, and thus formula (5) is transformed to:

$$\varepsilon_{p2}(k) = \varepsilon_p(k) + (1 - l_2(k))\pi_p(k) \tag{9}$$

where $\pi_p(k)$ denotes the gain parameter of the FDI attacks, oscillating randomly within a predefined boundary. The occurrence of an FDI attack is ascertained using $l_2(k)^* = \exp\{-\|\varepsilon_p(k) - \hat{\varepsilon}_p(k)\|\}$ If $l_2(k)^*$ is below a predefined positive threshold denoted by $\nu$, the attack is deemed to occur, resulting in $l_2(k) = 0$.

In the scenario where $i = 3$, the system faces Deception Attacks (DA), and equation (5) is modified to:

$$\varepsilon_{p3}(k) = (-1)^{1-l_3(k)}\varepsilon_p(k) \tag{10}$$

The variable $l_3(k)$ determines the success of the DA, conforming to a Bernoulli distribution. A value of $l_3(k) = 0$, signifies a successful attack, occurring with certainty $\mathbb{P}\{l_3(k) = 0\} = \bar{l}$. Conversely, when $l_3(k) = 1$, the offensives did not succeed, with the probability $\mathbb{P}\{l_3(k) = 1\} = 1 - \bar{l}$ for failure..

Ultimately, $l(k)$ signifies the outcome of a mixed attack, with $l(k) = 1$ representing a successful attack and $l(k) = 0$ indicating a failed attack. Here, $l(k) = l_1(k)l_2(k)l_3(k)$.

To conclude, the methodology presented advances a refined distributed output error formula that incorporates an element of predictive compensation:

$$\bar{\varepsilon}_p(k) = l(k)\varepsilon_p(k) + (1 - l(k))\hat{\varepsilon}_p(k) \tag{11}$$

where $\hat{\varepsilon}_p(k) = (\bar{\varepsilon}_p(k-1) + \hat{\Phi}_{p2}(k-1)\Delta x_p(k-1))$

*Remark 2:* Within this framework, DoS attacks are deliberate interruptions targeting network protocol implementation, aimed at rendering the computer or network incapable of providing standard services or accessing resources. FDI and DA attacks involve attackers injecting false signals to replace authentic information, thereby preventing the system from achieving its intended goals.

## B. Design of Second-order System

For subsystem 1, due to the compression of the nonlinear dynamic characteristics of the system into $\Phi_{p1}(k)$, obtaining its dynamic model remains challenging, but numerical variations can be estimated. Therefore, the cost function for $\Phi_{p1}(k)$ is given in the following form:

$$J(\hat{\Phi}_{p1}(k)) = |\Delta x_p(k+1) - \hat{\Phi}_{p1}(k)\Delta u_p(k-1)|^2 + \mu|\hat{\Phi}_{p1}(k) - \hat{\Phi}_{p1}(k-1)|^2$$

Taking the partial derivative of $\hat{\Phi}_{p1}(k)$ from the above equation yields 0, with

$$\hat{\Phi}_{p1}(k) = \hat{\Phi}_{p1}(k-1) + \frac{\eta\Delta u_p(k-1)}{\mu + |\Delta u_p(k-1)|^2} (\Delta x_p(k+1) - \hat{\Phi}_{p1}(k-1)\Delta u_p(k-1))$$

where $\eta \in (0, 2]$ is the step-size factor, which enhances the algorithm's flexibility and generality, and $\mu$ is a positive constant.

Likewise, in the case of subsystem 2, the associated cost function is formulated in the subsequent manner:

$$J(\hat{\Phi}_{p2}(k)) = |\Delta\varepsilon_p(k+1) - \hat{\Phi}_{p2}(k)\Delta x_p(k-1)|^2 + \mu|\hat{\Phi}_{p2}(k) - \hat{\Phi}_{p2}(k-1)|^2$$

Taking the partial derivative of $\hat{\Phi}_{p2}(k)$ from the above equation yields 0, with

$$\hat{\Phi}_{p2}(k) = \hat{\Phi}_{p2}(k-1) + \frac{\eta\Delta x_p(k-1)}{\mu + |\Delta x_p(k-1)|^2} (\Delta\varepsilon_p(k+1) - \hat{\Phi}_p(k-1)\Delta x_p(k-1))$$

Based on this, a control protocol is formulated for subsystem 1 and subsystem 2,

$$u_p(k) = u_p(k-1) + \frac{\rho\hat{\Phi}_{p1}(k)}{\lambda + \left|\hat{\Phi}_{p1}(k)\right|^2} (\hat{x}_p(k) - x_p(k))$$

$$x_p(k) = x_p(k-1) + \frac{\rho\hat{\Phi}_{p2}(k)}{\lambda + \left|\hat{\Phi}_{p2}(k)\right|^2}\bar{\varepsilon}_p(k)$$

Taking into account the current position and velocity state at the present time, the virtual desired velocity state is designed as follows:

$$\begin{cases} \hat{x}_p(k) = \hat{x}_p(k-1) + \frac{\rho\hat{\Phi}_{p1}^2(k)}{\lambda + \hat{\Phi}_{p1}^2(k)}\tilde{x}_p(k-1) + g \\ g = -\tilde{x}_p(k-1) \end{cases}$$

Consequently, the full control structure can be delineated below

$$\hat{\Phi}_{p1}(k) = \hat{\Phi}_{p1}(k-1) + \frac{\eta\Delta u_p(k-1)}{\mu + |\Delta u_p(k-1)|^2} (\Delta x_p(k) - \hat{\Phi}_{p1}(k-1)\Delta u_p(k-1)) \tag{12}$$

$$\hat{\Phi}_{p1}(k) = \hat{\Phi}_{p1}(1) \text{ if } \left|\hat{\Phi}_{p1}(k)\right| \leq \xi \text{ or } |\Delta u_p(k-1)| \leq \xi$$

$$u_p(k) = u_p(k-1) + \frac{\rho\hat{\Phi}_{p1}(k)}{\lambda + \left|\hat{\Phi}_{p1}(k)\right|^2}(\hat{x}_p(k) - x_p(k)) \quad (13)$$

$$\hat{\Phi}_{p2}(k) = \hat{\Phi}_{p2}(k-1) + \frac{\eta^*\Delta x_p(k-1)}{\mu^* + |\Delta x_p(k-1)|^2}(\Delta\bar{\varepsilon}_p(k) \\ - \hat{\Phi}_{p2}(k-1)\Delta x_p(k-1)) \quad (14)$$

$$\hat{\Phi}_{p2}(k) = \hat{\Phi}_{p2}(1) \text{ if } \left|\hat{\Phi}_{p2}(k)\right| \leq \xi^* \text{ or } |\Delta x_p(k-1)| \leq \xi^*$$

$$x_p(k) = x_p(k-1) + \frac{\rho\hat{\Phi}_{p2}(k)}{\lambda^* + \left|\hat{\Phi}_{p2}(k)\right|^2}\bar{\varepsilon}_p(k) \quad (15)$$

$$\begin{cases} \hat{x}_p(k) = \hat{x}_p(k-1) + \frac{\rho\hat{\Phi}_{p1}^2(k)}{\alpha + \hat{\Phi}_{p1}^2(k)}\tilde{x}_p(k-1) + g \\ g = -\tilde{x}_p(k-1) \end{cases} \quad (16)$$

The above MFAC consists of PPD estimation algorithm (12) (14), PPD reset algorithm, control algorithm (13) (15), and virtual estimation algorithm (16), which is designed with compensated distributed output $\Delta\bar{\varepsilon}_p(k)$.

### C. Convergence review

In this segment, we delve into the convergence properties of subsystem 1 when it is subjected to stochastic cyber-intrusions, with the principal outcomes encapsulated in Theorem 2.

*Theorem 2:* For subsystem 1 experiencing random cyber-attacks, provided assumptions 1-4 are fulfilled and the controller parameters are selected as $\lambda > 0$, $0 < \eta < 2$, $\mu > 0$, $\rho \in (0,1]$, The control methodologies outlined in equations (12) to (16) are instrumental in attaining this consensus.

*Proof:* The proof is structured into four main parts. First, it is verified that the PPD parameter approximation error is bounded. Second, we show the boundedness of the error in virtual tracking speed. Third, the verification of the distributed output forecast error's bounded nature is addressed. Fourth, the argument concludes with the proof of the distributed output error's boundedness. The rationale behind these bounded attributes is rooted in the mathematical expectation principle.

*part-1:* Let $\tilde{\Phi}_{p1}(k) = \hat{\Phi}_{p1}(k) - \Phi_{p1}(k)$ denote the approximation error of the PPD parameter, as stated in (12).

$$\tilde{\Phi}_{p1}(k) = \left(1 - \frac{\eta\Delta u_p(k-1)^2}{\mu + |\Delta u_p(k-1)|^2}\right) \\ \times \tilde{\Phi}_{p1}(k-1) + \Phi_{p1}(k-1) - \Phi_{p1}(k). \quad (17)$$

From equation (20), it is derived that

$$|\tilde{\Phi}_{p1}(k)| \leq \left|\left(1 - \frac{\eta\Delta u_p(k-1)^2}{\mu + |\Delta u_p(k-1)|^2}\right)\right||\tilde{\Phi}_{p1}(k-1)| \\ + |\Phi_{p1}(k-1) - \Phi_{p1}](k)|. \quad (18)$$

Since $|\Delta u_p(k)| \leq b$, by appropriately selecting $\eta$ and $\mu$ such that $0 < \eta \leq 1$ and $\mu \geq 0$, there exists a constant $q_1$ ensuring that holds.

$$0 < \left(1 - \frac{\eta\Delta u_p(k-1)^2}{\mu + |\Delta u_p(k-1)|^2}\right) \leq q_1 < 1 \quad (19)$$

Since $|\Phi_{p1}(k)| \leq a$, in light of Assumption 4, it follows that the difference $|\Phi_{p1}(k-1) - \Phi_{p1}(k)| \leq a$.

Drawing from equations (8) and (9), we deduce that

$$|\tilde{\Phi}_{p1}(k)| \leq q_1|\tilde{\Phi}_{p1}(k-1)| + a \\ \leq \cdots \\ \leq q_1^k|\hat{\Phi}_{p1}(0)| + \frac{a(1-q_1^k)}{1-q_1} \quad (20)$$

which implies that $\tilde{\Phi}_{p1}(k)$ is bounded. Consequently, $\Phi_{p1}(k)$ is also bounded as $\tilde{\Phi}_{p1}(k)$ is bounded.

*part-II:* Let $\tilde{\Phi}_{p2}(k) = \hat{\Phi}_{p2}(k) - \Phi_{p2}(k)$ be the estimation discrepancy associated with the PPD parameter, as per equation (14)

$$\tilde{\Phi}_{p2}(k) = \tilde{\Phi}_{p2}(k-1) - \Delta\Phi_{p2}(k) + \frac{\eta^*\Delta x_p(k-1)}{\mu^* + |\Delta x_p(k-1)|^2} \\ (\bar{\varepsilon}_p(k) - \bar{\varepsilon}_p(k-1) - \hat{\Phi}_{p2}(k-1)\Delta x_p(k-1)) \\ = \tilde{\Phi}_{p2}(k-1) - \Delta\Phi_{p2}(k) + \frac{\eta^*\Delta x_p(k-1)}{\mu^* + |\Delta x_p(k-1)|^2} \\ (l(k)\varepsilon_p(k) - l(k)\hat{\varepsilon}_p(k)) \\ = [1 - \frac{\eta^*\Delta x_p^2(k-1)l(k)}{\mu^* + |\Delta x_p(k-1)|^2}]\tilde{\Phi}_{p2}(k-1) - \Delta\Phi_{p2}(k) \quad (21)$$

By evaluating the absolute value and then employing the expectation operator across the entirety of the preceding equation, the result is

$$\mathbb{E}\{|\tilde{\Phi}_{p2}(k)|\} \leq |1 - \frac{\eta^*\Delta x_p^2(k-1)l(k)}{\mu^* + \Delta x_p^2(k-1)}|\mathbb{E}\{|\tilde{\Phi}_{p2}(k-1)|\} \\ + |\Delta\Phi_{p2}(k)| \quad (22)$$

Observe that for $0 < \eta^* < 2$, $\mu^* > 0$, $0 < l(k) < 1$. Consequently, a positive value $d$ is identified, fulfilling the condition that $0 < |1 - \frac{\eta^*\Delta x_p^2(k-1)l(k)}{\mu^* + \Delta u^2(k-1)}| = d < 1$. Also because of $|\Phi_{p2}(k)| \leq \mathfrak{b}_2$, so $|\Delta\Phi_{p2}(k)| \leq 2\mathfrak{b}_2$. Consequently, the formulation of equation (22) is articulated to be,

$$\mathbb{E}\{|\tilde{\Phi}_{p2}(k)|\} \leq d\,\mathbb{E}\{|\tilde{\Phi}_{p2}(k-1)|\} + 2\mathfrak{b}_2 \\ \leq \cdots \\ \leq d^{k-1}\mathbb{E}\{|\tilde{\Phi}_{p2}(1)|\} + \frac{2\mathfrak{b}_2}{1-d} \quad (23)$$

Based on the inequality above, it is evident that $\tilde{\Phi}_{p2}(k)$ is uniformly bounded. Therefore, based on $\Phi_{p2}(k)$ being bounded, $\hat{\Phi}_{p2}(k)$ is also bounded.

*part-III:* Define the virtual desired velocity tracking error $\tilde{x}_p(k) = \hat{x}_p(k) - x_p(k)$. Utilizing equations (15) and (16), may be represented in the form of

$$\tilde{x}_p(k+1) = \hat{x}_p(k+1) - \hat{x}_p(k) + \hat{x}_p(k+1) - x_p(k+1)$$
$$= \Delta\hat{x}_p(k+1) + (1 - \frac{\rho\hat{\Phi}_{p1}(k)\Phi_{p1}(k)}{\alpha + \hat{\Phi}_{p1}^2(k)})\tilde{x}_p(k) \tag{24}$$

Let a Lyapunov function be defined: $v(k) = \tilde{x}_p^2(k)$

$$\Delta v(k+1) = (\tilde{x}_p(k+1) - \tilde{x}_p(k))(\tilde{x}_p(k+1) + \tilde{x}_p(k))$$
$$= 2\tilde{x}_p(k)(\Delta\tilde{x}_p(k+1) - \frac{\rho\hat{\Phi}_{p1}(k)\Phi_{p1}(k)}{\alpha + \hat{\Phi}_{p1}^2(k)}\tilde{x}_p(k))$$
$$(\Delta\tilde{x}_p(k+1) - \frac{\rho\hat{\Phi}_{p1}(k)\Phi_{p1}(k)}{\alpha + \hat{\Phi}_{p1}^2(k)}\tilde{x}_p(k))^2 \tag{25}$$

when $\lim_{k\to\infty} a = 0$, so $\lim_{a\to 0 \, k\to\infty} \tilde{\Phi}_{p1}(k) = 0$

$$\Delta v(k+1) = 2\tilde{x}_p(k)(g + \frac{\rho\hat{\Phi}_{p1}(k)\tilde{\Phi}_{p1}(k)}{\alpha + \hat{\Phi}_{p1}^2(k)}\tilde{x}_p(k))$$
$$(g + \frac{\rho\hat{\Phi}_{p1}(k)\tilde{\Phi}_{p1}(k)}{\alpha + \hat{\Phi}_{p1}^2(k)}\tilde{x}_p(k))^2$$
$$= g^2 + 2\tilde{x}_p(k)g$$
$$= -\tilde{x}_p^2(k) \tag{26}$$

Evidently, with the control scheme delineated within this document, the condition $\Delta v(k) < 0$ is reliably met. The divergence between the current velocity and the envisioned target velocity achieves asymptotic and sustained convergence, thereby finalizing the verification.

*part-IV:* Let us denote the distributed output forecast discrepancy by $\bar{\varepsilon}_p(k) = \hat{\varepsilon}_p(k) - \varepsilon_p(k)$. Utilizing the formulations from equation (11), this discrepancy is articulated in the subsequent manner:

$$\tilde{\varepsilon}_p(k) = \hat{\varepsilon}_p(k) - \varepsilon_p(k) \tag{27}$$
$$= \bar{\varepsilon}_p(k-1) + \hat{\Phi}_{p2}(k-1)\triangle x_p(k-1)$$
$$- \varepsilon_p(k-1) - \Phi_{p2}(k-1)\Delta x_p(k-1)$$
$$= (1 - l(k))\tilde{\varepsilon}_p(k-1) + \hat{\Phi}_{p2}(k-1)$$
$$\triangle x_p(k-1) \tag{28}$$

Obtain the absolute magnitude and then employ the expected value operator on the expression designated as equation (27), and scale them to derive the result, then

$$\mathbb{E}\{|\tilde{\varepsilon}_p(k)|\} \leq (1 - l(k))\mathbb{E}\{|\tilde{\varepsilon}_p(k-1)|\}$$
$$+ |\tilde{\Phi}_{p2}(k-1)\triangle x_p(k-1)| \tag{29}$$

Due to the boundedness of $\hat{\Phi}_{p1}(k)$ proven in the part-I, and from our findings in the part-II that $x_p(k-1)$ is also bounded, we can conclude that $\hat{\Phi}_{p1}(k)x_p(k-1)$ is bounded.

Let $\hat{\Phi}_{p1}(k)x_p(k-1) < q$, where $q$ is a positive constant. Given that $l \in (0,1)$, the relational expression is subsequently reformulated to read:

$$\mathbb{E}\{|\tilde{\varepsilon}_p(k)|\} \leq (1-l)\mathbb{E}\{|\tilde{\varepsilon}_p(k-1)|\} + q \tag{30}$$
$$\leq \cdots$$
$$\leq (1-l)^{k-1}\mathbb{E}\{|\tilde{\varepsilon}_p(1)|\} + \frac{q}{1-l} \tag{31}$$

This indicates that the value of $\tilde{\varepsilon}_p(k)$ remains consistently confined within limits.

*part-V:* Due to (14) and $e_{\varepsilon_p}(k+1) = \varepsilon_p^* - \varepsilon_p(k-1)$, $e_{\varepsilon_p}(k+1)$ can be expressed as,

$$e_{\varepsilon_p}(k+1) = \varepsilon_p^* - \varepsilon_p(k-1)$$
$$= e_{\varepsilon_p}(k) - \Delta\varepsilon_p(k-1)$$
$$= (1 - \frac{\rho^*\hat{\Phi}_{p2}(k)\Phi_{p2}(k)}{\lambda^* + \hat{\Phi}_{p2}^2(k)})e_{\varepsilon_p}(k)$$
$$+ \frac{\rho^*\hat{\Phi}_{p2}(k)\Phi_{p2}(k)}{\lambda^* + \hat{\Phi}_{p2}^2(k)}\hat{\varepsilon}_p(k) \tag{32}$$

let $\frac{\rho^*\hat{\Phi}_{p2}(k)\Phi_{p2}(k)}{\lambda^* + \hat{\Phi}_{p2}^2(k)} = m(k)$

Employing an identical approach as outlined in part-I, equation (35) is subsequently reformulated in the following manner:

$$|e_{\varepsilon_p}(k+1)| \leq |1 - m(k)| |e_{\varepsilon_p}(k)|$$
$$|m(k)| |l(k)\tilde{\varepsilon}_p(k) + \tilde{\Phi}_{p2}(k-1)\Delta x_p(k-1)| \tag{33}$$

Based on the evidence presented within the demonstrations of part I and II, $\tilde{\varepsilon}_p(k)$ and $\tilde{\Phi}_{p2}(k-1)$ are bounded, so let $(1-D)l\mathbb{E}\{|\tilde{\varepsilon}_p(k)|\} + (1-D)\mathbb{E}\{|\tilde{\Phi}_{p2}(k-1)\Delta x_p(k-1)|\} < d^*$ with $d^* > 0$ being a constant.

Consequently, equation (29) is articulated in the subsequent format:

$$\mathbb{E}\{|e_{\varepsilon_p}(k+1)|\} \leq D\mathbb{E}\{|e_{\varepsilon_p}(k)|\} + (1-D)l\mathbb{E}\{|\tilde{\varepsilon}_p(k)|\}$$
$$(1-D)\mathbb{E}\{|\tilde{\Phi}_{p2}(k-1)\Delta x_p(k-1)|\}$$
$$\leq D\mathbb{E}\{|e_{\varepsilon_p}(k)|\} + d^*$$
$$\leq D^k\mathbb{E}\{|e_{\varepsilon_p}(1)|\} + \frac{d^*}{1-D} \tag{34}$$

which illustrate $\varepsilon_p^*$ is bounded. Due to $e_{\varepsilon_p}(k+1)$ is bounded, $\varepsilon_p(k-1)$ is also bounded.

The conclusion of the demonstration has been reached, this crucial result ensures that the proposed control algorithm effectively maintains system stability and achieves its intended control objectives under the specified conditions. The meticulous application of Lyapunov functions and the detailed step-by-step analysis provide a solid theoretical foundation for our approach. This comprehensive proof not only validates the robustness and reliability of the proposed control strategy but also highlights its practical applicability in real-world scenarios where multi-agent systems are subject to mixed network attacks. Consequently, we can confidently assert that the control strategy developed in this paper is both sound

and effective, offering significant potential for enhancing the performance and resilience of multi-agent systems in dynamic and potentially hostile environments.

## IV. SIMULATION

In this segment, we introduce a computational example to substantiate the efficacy of the suggested methodology.

*Example 1:* Consider the MASs comprising a single leader and four subordinate agents, as depicted in the communication structure diagram in Fig. 1.

The dynamics of each constituent agent's model are delineated below:

$$
\begin{cases}
Agent1 : y_1(k+1) = \dfrac{y_1(k)x_1(k)}{1+y_1^2(k)} + \dfrac{x_1(k)u_1(k)}{1+x_1^2(k)} + u_1(k) \\[2mm]
Agent2 : y_2(k+1) = \dfrac{y_2(k)x_2(k)}{1+y_2^2(k)} + \dfrac{x_2(k)u_2(k)}{1+x_2^3(k)} + 0.5u_2(k) \\[2mm]
Agent3 : y_3(k+1) = \dfrac{y_3(k)x_3(k)}{1+y_3^2(k)} + \dfrac{x_3(k)u_3(k)}{1+x_3^2(k)} + 0.9u_3(k) \\[2mm]
Agent4 : y_4(k+1) = \dfrac{y_4(k)x_4(k)}{1+y_4^2(k)} + \dfrac{x_4(k)u_4(k)}{1+x_4^5(k)} + 0.8u_4(k)
\end{cases}
$$

Furthermore, one can determine the Laplacian matrix from the interaction structure diagram displayed in Fig. 2, as follows:

$$
\mathcal{L} = \begin{bmatrix} 1 & 0 & 0 & -1 \\ 0 & 1 & -1 & 0 \\ 0 & -1 & 2 & -1 \\ 0 & -1 & -1 & 2 \end{bmatrix}
$$

The trajectory of the leading agent is defined through the subsequent mathematical expression:

$$
y_d(k) = \begin{cases} 2, & 0 < k \le 200 \\ 0.5, & 200 < k \le 400 \end{cases}
$$

Here , the starting values and controller parameters for the agents are: $\bar{l}_1 = 0.5$, $\bar{l}_2 = 0.55$, $\bar{l}_3 = 0.45$, $\bar{l}_4 = 0.55$ and The gain parameter for FDI attacks $\pi_p(k)$ varies within the spectrum encompassing $[0, 5]$. The starting parameters and regulatory configurations are outlined below $y_p(0) = [1; 1; 1; 1]$, $u_p(0) = [1; 1; 1; 1]$, $\hat{\Phi}_{p1}(k) = [1.05; 1.1; 1.2; 1.03]$, $\hat{\Phi}_{p2}(k) = [1.2; 1.1; 1.02; 1.3]$, $\xi = 10^{-5}$ , $\rho = 0.3$, $\eta = 1.5$, $\lambda = 4$, $\mu = 0.5$, $p = 1, 2, 3, 4.\rho^* = 0.3$, $\eta^* = 1.5$, $\lambda^* = 4$, $\mu^* = 0.5$, $p = 1, 2, 3, 4$. The outcomes of the simulation are depicted within Fig. 3 to Fig. 6. Fig. 3 illustrates the tracking performance under a time-invariant signal in Example 1, comparing the proposed algorithm, a reference algorithm, and the performance without compensation. Fig. 5 illustrates the tracking performance under a time-varying signal in Example 2, using the proposed algorithm, a comparative algorithm, and an uncompensated scenario. Fig. 4 . 6 . 7 and 8 present the error trajectories for the systems under the proposed and comparative algorithms. This clear comparison demonstrates that the proposed method yields good control effects in simulations.

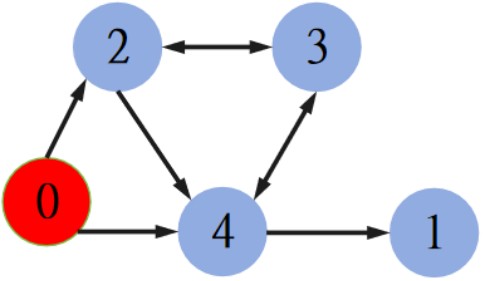

Fig. 2. Structural diagram of the Examples 1 and 2.

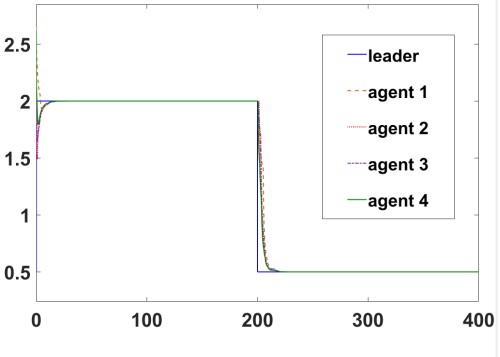

Time(k)

(a)

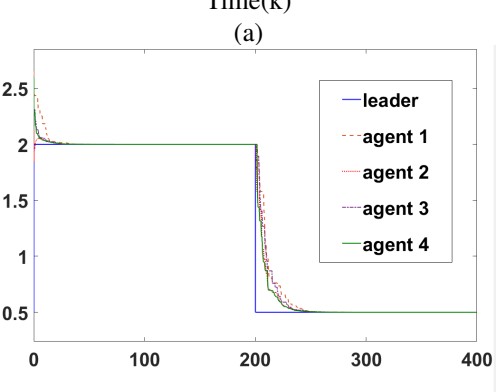

Time(k)

(b)

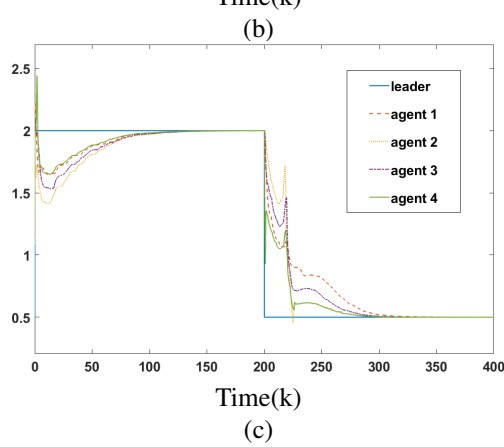

Time(k)

(c)

Fig. 3. Performance of multi-agent systems tracking under mixed attacks. (a) Proposed MFAC algorithm. (b) Control methodology described in reference [24]. (c) Control strategy without any compensation.

*Remark 4:* In the course of the simulation, the predetermined path for the fictitious leader is designated at node O. In this scenario, only agents 2 and 4 directly receive information from the leader, while agents 1 and 3 receive information indirectly through agents 2 and 4.

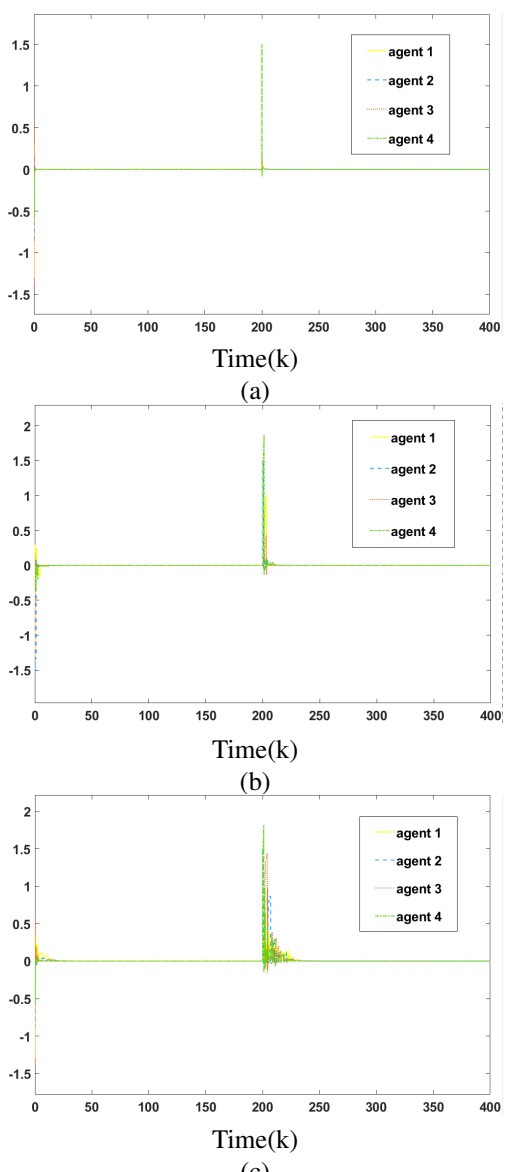

Fig. 4. The tracking error of multi-agent systems under mixed attacks. (a) Proposed MFAC algorithm. (b) Control methodology described in reference [24]. (c) Control strategy without any compensation.

The aforementioned simulation results indicate that, under time-invariant signals, the proposed MFAC algorithm exhibits superior control performance compared to the control method described in reference [24]. Specifically, the proposed algorithm achieves smaller errors and higher stability. Furthermore, the compensation method introduced in this paper demonstrates a significant improvement over the scenario without compensation. The control effect with compensation is markedly better than that without compensation. In summary, the MFAC

algorithm with compensation proposed in this paper shows excellent control performance under time-invariant signals, making it a robust and effective solution for managing system stability and accuracy.

*Example 2:* The multi-agent systems with a time-varying signal is analyzed, as showcased by the connectivity diagram in Fig. 1.

The path followed by the leader is defined by the formula below:

$$y_d(k) = 0.6 + 0.2(\sin(\frac{2\pi k}{50}) + \sin(\frac{2\pi k}{100}) + \sin(\frac{2\pi k}{150}))$$

Here, the starting conditions and controller settings for the agents are: $\bar{l}_1 = 0.45$, $\bar{l}_2 = 0.6$, $\bar{l}_3 = 0.55$, $\bar{l}_4 = 0.5$ and The gain parameter for FDI attacks $\pi_p(k)$ varies randomly between $[0, 6]$. The starting states and the parameters for regulation are set out below $y_p(0) = [0; 0; 0; 0]$, $u_p(0) = [1; 1; 1; 1]$, $\hat{\Phi}_{p1}(0) = [1.05; 1.1; 1.02; 1.3]$, $\hat{\Phi}_{p2}(0) = [1.2; 1.1; 1.02; 1.3]$, $\xi = 10^{-5}$, $\rho = 0.3$, $\eta = 1.5$, $\lambda = 4$, $\mu = 0.5$, $p = 1, 2, 3, 4.\rho^* = 0.3$, $\eta^* = 1.5$, $\lambda^* = 4$, $\mu^* = 0.5$, $p = 1, 2, 3, 4.$ as illustrated by the outcomes presented within the simulation Fig. 4.Simulation results demonstrate that, whether under time-varying or time-invariant signals, the proposed algorithm exhibits smaller errors, faster convergence, and better tracking performance compared to the algorithm presented in [24]. Through clear comparisons across various aspects, the proposed method shows good control effects in simulations. The simulation results illustrated above indicate that, under time-varying signals, the proposed MFAC algorithm demonstrates markedly better control performance compared to the control strategy outlined in reference [24]. This improvement is reflected in significantly reduced error margins and enhanced system stability. Moreover, the introduction of the compensation method proposed in this study shows a clear and substantial improvement over the scenario without compensation. Specifically, the control effect achieved with compensation is considerably superior to that without compensation. Thus, it can be concluded that the MFAC algorithm, when coupled with the proposed compensation technique, delivers outstanding control performance under time-varying signal conditions. This highlights the algorithm's robustness and efficacy in managing dynamic systems with fluctuating signals, ensuring both precision and stability in control outcomes.

## V. CONCLUSION

The present study investigates the adaptive control challenge for second-order nonlinear multi-agent networks subjected to a combination of cyber threats. To facilitate ongoing scholarly work, the output discrepancies across agents within each subsystem are determined using backstepping methodologies devoid of model dependencies, subsequently converting them into a linear form through dynamic linearization. This technique streamlines the control formulation, making it more manageable. Additionally, the paper delves into the orchestration of hybrid cyber-assaults, proposing a countermeasure mechanism designed to alleviate their detrimental effects. Such

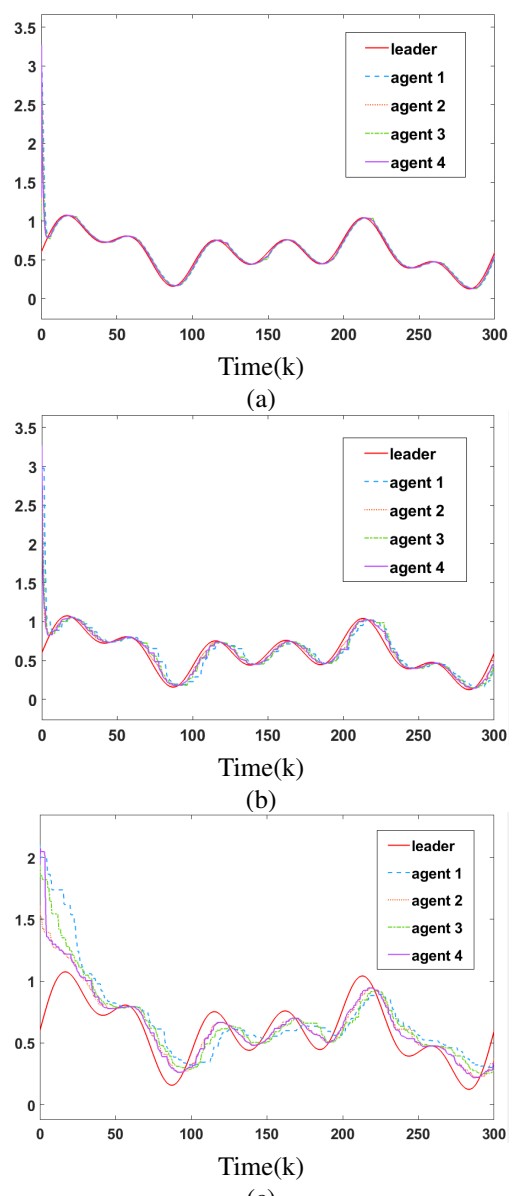

Fig. 5. Tracking performance of multi-agent systems under mixed attacks. (a) The proposed MFAC algorithm. (b) The control algorithm in reference [24]. (c) The control algorithm without compensation.

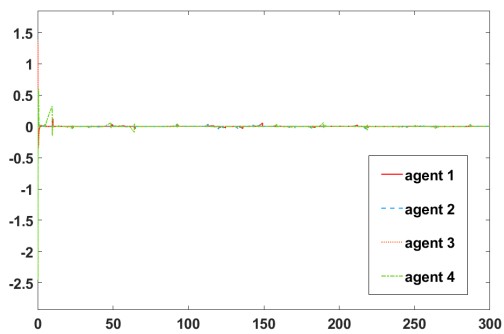

Fig. 6. Error trajectories $e_{\varepsilon_p}(k)(p = 1, 2, 3, 4)$ of the proposed algorithm.

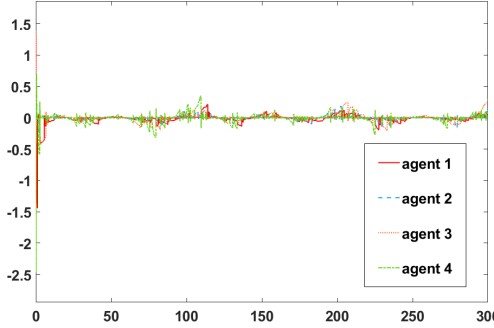

Fig. 7. Error trajectories $e_{\varepsilon_p}(k)(p = 1, 2, 3, 4)$ of the algorithm proposed in reference [24].

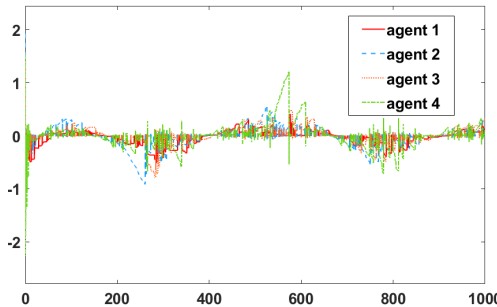

Fig. 8. Error trajectories $e_{\varepsilon_p}(k)(p = 1, 2, 3, 4)$ of the algorithm without compensation.

a mechanism enhances system resilience to diverse cyber-intrusions. The convergence of the strategy is substantiated through the application of Lyapunov stability theory, establishing a robust theoretical foundation. In conclusion, empirical simulations corroborate the commendable control efficacy of the method, preserving system steadiness and potency amidst adverse scenarios.

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
