# OpenReview forum: "Model-Free Adaptive Control of Second-Order Multi-Agent Systems via Backstepping Under Mixed Attacks"
_IEEE.org/ICIST/2024/Conference — IEEE ICIST 2024 Conference Submission_

### Official Review · Reviewer_Qf35 · 2024-08-22
**Model-Free Adaptive Control of Second-Order Multi-Agent Systems via Backstepping Under Mixed Attacks**

**Rating:** 6
**Confidence:** 5

**Review:**

This paper employs model-free adaptive control methods to investigate second-order multi-agent systems under mixed attacks. This work is well organized. Below are some comments.
(1)	The contributions should be illustrated in a clearer manner. For example, what is the main improvement of the paper compared to the existing results. The authors should explain the unique contributions of this paper.
(2)	I think that adding a block diagram/scheme will help to understand how your method works.
(3)	Is the computational complexity of the proposed algorithm too large?
(4)	The simulation results should be explained more carefully.
(5)    The quality of Figures should be improved.
(6)	The paper is well presented, spelled correctly. I recommend authors to carefully read the entire paper to find possible misspellings.

---

### Official Review · Reviewer_NaBd · 2024-08-24
**Review Comments for Manuscript No. 138**

**Rating:** 7
**Confidence:** 4

**Review:**

1. The manuscript contains several noticeable errors in details such as text formatting, figure text size, and reference formatting. The authors should correct these issues.

2. Given that the study focuses on second-order systems, what are the advantages of this work compared to previous studies on strict-feedback MAS systems based on the backstepping method? In other words, why can't previous research be directly extended to the current study?

3. Regarding the cost function introduced after equation (11), what is the basis for choosing its particular form?

4. The physical meaning or the actual significance of the state vector in equation (1) is not clearly defined. However, the authors mention in the text before equation (12), “Taking into account the current position and velocity state at the present time,” which is confusing.

5. The simulation section needs improvement in the following aspects: First, in the Example, the form of the agents provided by the authors does not match the second-order agents described in equation (1). Additionally, the specific dynamics of the network attacks are not presented.

---

### Official Review · Reviewer_wETr · 2024-08-24
**This article is very interesting and a good one.**

**Rating:** 7
**Confidence:** 5

**Review:**

In this paper,  the model-free adaptive control methods were proposed to investigate the second-order multi-agent systems under
mixed attacks. The obtained result is valuable and can be accepted if the following problems can be clarified.
1. In this paper, the control problem of second-order multi-agent systems under mixed attacks is studied. Can the method proposed in this paper be extended to solve the control problem of high order multi-agent systems?
2. In the introduction, there is a lack of relevant research on the control methods of multi-agent systems under mixed attack.
3. Please explain why (1) is a second-order system?

---

### Decision · Program_Chairs · 2024-09-08

Accept (Oral)